# Bone Remodeling Interaction with Magnesium Alloy Implants Studied by SEM and EDX

**DOI:** 10.3390/ma15217529

**Published:** 2022-10-27

**Authors:** Alexey Drobyshev, Alexander Komissarov, Nikolay Redko, Zaira Gurganchova, Eugene S. Statnik, Viacheslav Bazhenov, Iuliia Sadykova, Andrey Miterev, Igor Romanenko, Oleg Yanushevich

**Affiliations:** 1Laboratory of Medical Bioresorption and Bioresistance, Moscow State University of Medicine and Dentistry, 127473 Moscow, Russia; 2Laboratory of Hybrid Nanostructured Materials, National University of Science and Technology “MISiS”, 119049 Moscow, Russia; 3HSM Laboratory, Center for Energy Science and Technology, Skoltech, 121205 Moscow, Russia; 4Casting Department, National University of Science and Technology “MISiS”, 119049 Moscow, Russia

**Keywords:** bioresorption, magnesium alloys, bioresorbable materials

## Abstract

The development direction of bioresorbable fixing structures is currently very relevant because it corresponds to the priority areas in worldwide biotechnology development. Magnesium (Mg)-based alloys are gaining high levels of attention due to their promising potential use as the basis for fixating structures. These alloys can be an alternative to non-degradable metal implants in orthopedics, maxillofacial surgery, neurosurgery, and veterinary medicine. In our study, we formulated a Mg-2Zn-2Ga alloy, prepared pins, and analyzed their biodegradation level based on SEM (scanning electron microscopy) and EDX (energy-dispersive X-ray analysis) after carrying out an experimental study on rats. We assessed the resorption parameters 1, 3, and 6 months after surgery. In general, the biodegradation process was characterized by the systematic development of newly formed bone tissue. Our results showed that Mg-2Zn-2Ga magnesium alloys are suitable for clinical applications.

## 1. Introduction

The use of magnesium alloys as a basis for manufacturing biodegradable elements in medicine is promising [1]. Magnesium alloys have a high specific strength, and their density is similar to human bone [2]. In modern clinical practice, substances for manufacturing fixing and supporting structures for the human body include bone grafting materials, titanium alloys, cobalt–chromium, stainless steel, etc. [3,4]. However, allergic or inflammatory reactions are possible in the body when they are used because of their foreign nature. Using non-resorbable structures in pediatric practice is restricted because of child-body growth and the need to replace supporting elements when making a prosthetic appliance, especially in the temporomandibular joint [5]. Removing non-resorbable elements or revising the area in question entails traumatic operations associated with pain, a decrease in life quality, and an increase in treatment costs [6]. However, using magnesium alloy-based materials would solve these problems [2,7]. Magnesium alloys are preferred to existing resorbable systems because of their high Young’s modulus values and greater elasticity compared with polymer screws.

However, the clinical use of magnesium alloys in the human body may be limited due to inhomogeneous biodegradation, a high corrosion rate, and hydrogen release in the first 2–4 months, which may adversely affect the healing process [8,9,10,11]. Composition modification and alloy surface treatment can be used to solve these problems [7,11,12]. There is a significant number of studies devoted to additionally alloying magnesium by rare earth or other metallic elements to improve corrosion resistance [13,14,15].

Previously, Mg-Zn-Ga alloys were proposed as materials for osteosynthesis applications [16]. Zn and Ga exhibit the same high solution strengthening effect on Mg due to their similar atomic radii and high solubility in Mg [17,18]. Mg alloys can obtain considerable mechanical properties upon Zn and Ga additions after deformation processing [17,18,19]. Ga’s effectiveness in treating disorders associated with accelerated bone loss, and its antibacterial properties, underline the potential of Ga as an alloying element for biodegradable Mg alloys [20,21,22,23,24].

Our aim in this work was to record the resorption levels of fixing elements based on magnesium alloys in the body of a rat. In our experiment, we used energy-dispersive X-ray spectroscopy and scanning electron microscopy to quantitatively and qualitatively determine the distribution of elements in the screw implantation bed.

## 2. Materials and Methods

### 2.1. Biomaterial Preparation

To prepare the alloy of Mg—2 wt.% Zn—2 wt.% Ga, we used high-purity metals: magnesium Mg (99.95 wt.%), Zn (99.995 wt.%), and Ga (99.9999 wt.%). We carried out melting in a resistance furnace in a steel crucible coated with BN in a protective atmosphere of Ar + 2 vol.% SF_6_. We purged the resulting melt with Ar at 730–750 °C for 3 min and held it for 10 min before pouring it into a mold. Next, we cast an ingot with a diameter of 60 mm and a height of 200 mm into an aluminum mold. We heat treated the ingot: 300 °C for 15 h + 400 °C for 30 h. We machined a 145 mm-high and 50 mm-in-diameter cylindrical billet from the ingot. Then, we subjected the billet to hot extrusion on a vertical hydraulic press with a pressing force of 300 tons using the direct extrusion method at a speed of 1 mm/s and an extrusion ratio of 6. We obtained a cylindrical extruded rod with a diameter of 20 mm and a length of ~1 m. The temperatures of the die and the extruded billet were 200 and 150 °C, respectively. We cut pins with a diameter of 1.5 mm and a height of 5 mm from the rod using electro-erosion cutting. We cleaned the pins’ surfaces with emery bars.

### 2.2. Surgical Procedure

We conducted in vivo experiments in compliance with the rules of ethics and animal welfare. We used both sexes of Wistar-line white laboratory rats from 6 months old with average body weights of 340–400 g. The study protocol was approved by the interuniversity ethics committee (No. 04 of 15 April 2021), and also complied with the principles of the “European Convention for the Protection of Vertebrate Animals Used for Experiments or for Other Scientific Purposes” dated 18 March 1986.

We performed the operation under general anesthesia with an intramuscular injection of Flexoprofen (2.5% 10 mg per kg; Vic, Belarus) and Zoletil (20 mg per kg; Virbac, France). We used Brilocaine (1:200,000; Ferein, Russia) for local anesthesia. Under the conditions of the experimental operating room and in compliance with the rules of asepsis and antisepsis, we performed a skin incision in the femur area from the outside and isolated the bone. Each animal underwent an installation of three 1.5 mm-diameter and 5 mm-long femur-body implants. We sutured the wound in layers with Vicryl 4-0 (Ethicon, Raritan, NJ, USA). We treated the postoperative area with an antibacterial aerosol Terramycin (Zoetis, Germany). We carried out postoperative antibiotic therapy using an intramuscular injection of Convenia (Zoetis, Italy). The experiment duration was 6 months. We bred animals at 1, 3, and 6 months after surgery by intramuscular injection and overdose of Telazol (Zoetis, Parsippany-Troy Hills, NJ, USA).

### 2.3. Bioresorption Monitoring

We used a scanning electronic microscope (SEM) and energy-dispersive X-ray spectroscopy (EDX) to study the microstructure of the installed implants. We carried out the SEM study using an electron microscope VEGA 3 LMU (TESCAN ORSAY HOLDING, Brno, Czech Republic). We carried out energy-dispersive X-ray spectroscopy using an Oxford Instruments spectrometer integrated into the TESCAN microscope and AZtec control program. All images obtained with SEM are in secondary electrons. The signal of secondary electrons is sensitive to the topography of the sample surface, so we used the SE detectors (secondary electrons) when studying surface morphology. For example, to understand the general appearance of biological samples, an SE detector is needed to observe their fractures, pores and surface roughness.

We carried out biopsy preparation before our study. First, we cleaned by freeze-drying. This is a dehydration process used to maintain the physical and biological integrity of biological materials after storage for months or even years. The lyophilization process is based on freezing the material, which is followed by a decrease in external pressure, thereby allowing water to directly pass from a solid to a gaseous state. We deposited a thin conductive gold film on the surface of a nonconductive sample. This is necessary for obtaining better SEM images because non-conductive samples are charged when scanned with an electron probe, which overexposes the image.

We calibrated the EDX detector before specimen studies by standard reference material (SRM) 2910b obtained from the National Institute of Standards and Technology (NIST). Usually, this calibrant is used for evaluating the physical and chemical properties of HAp with biological, geological, or synthetic origins [25]. The Ca/P molar ratio of a SRM 2910b is 1.67, which corresponds to the theoretical Ca/P value of HAp with a composition of Ca_10_(PO_4_)_6_(OH)_2_. We extracted the quantitative information of elements distribution at locations where the bone surface was appreciably flat.

## 3. Results

### 3.1. Scanning Electron Microscopy (SEM) of Biopsy Preparations

We captured all SEM images with corresponding elemental maps near the location where the pin was implanted. For this reason, the results are comparable. We paid special attention to the area of bone-to-implant contact when analyzing SEM data. After 1 month, the implant is visualized, and is partially covered with the products of the interaction of its resorption and bone tissue (the so-called “interface”) (Figure 1). However, on the implant’s right side, there are fewer of these products; therefore, a surface more similar to the original implant is visible. A layer of corrosion products is formed on the surface of the implant as a result of the interactions between body fluids and implant metals. The formed layer of corrosion products gradually dissolves in body fluids (Figure 2). A larger layer of bone–implant junctions exists on the left side of the implant specimen; therefore, these sites were closer in appearance to the original bone.

The physical bioresorption of the implant occurred after 3 months, and it was impossible to visualize it after this period. The implant remained within the bone and we found no loss of stability. (Figure 3). We observed pores and spherical discharges in the proposed installation site (Figure 4).

**Figure 3 materials-15-07529-f003:**
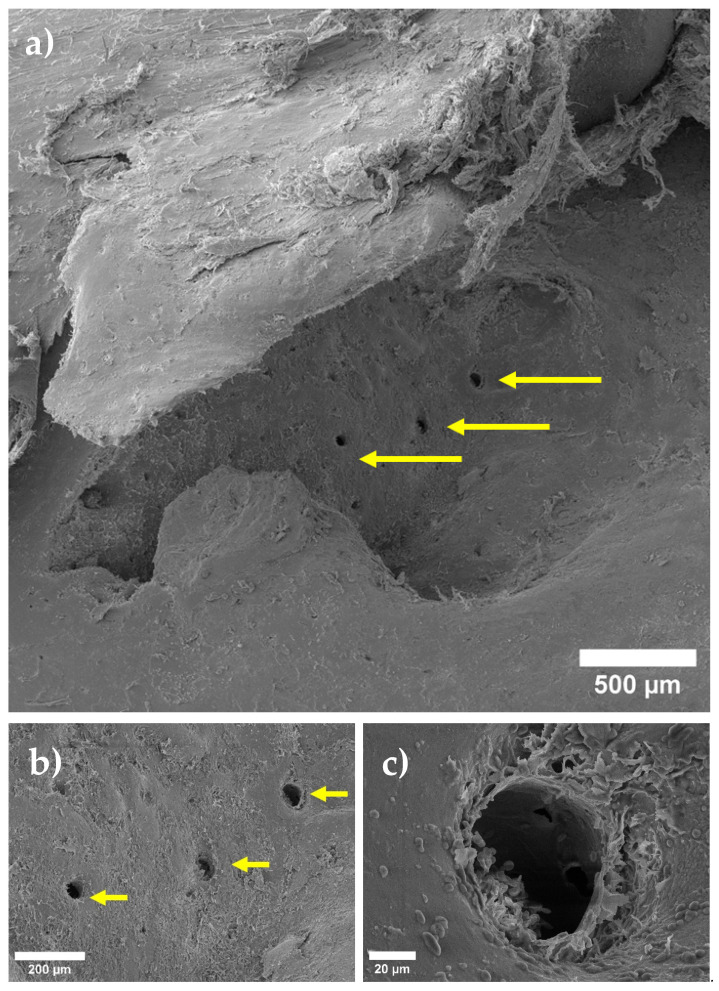
SEM image of bone area after 3 months at different magnifications: (**a**) small, (**b**) medium, and (**c**) large, respectively. Arrows in picture show pores associated with release of hydrogen during biodegradation of magnesium alloy.

**Figure 4 materials-15-07529-f004:**
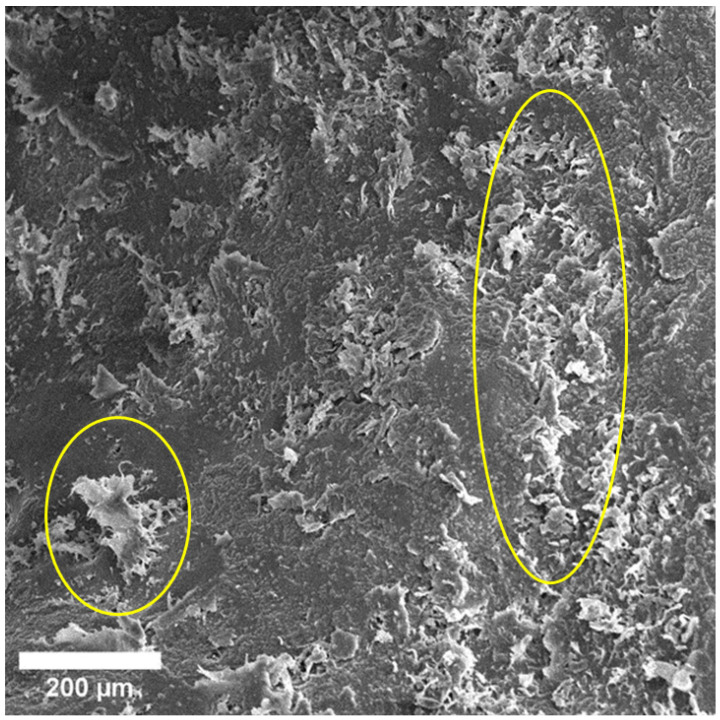
SEM image of bone area at proposed implant site. Spherical discharges are determined on surface of bone tissue. Similar results are found with a bone sample at 6 months (Figure 5). There are no visible traces of implant at the location of implant, and spherical discharges are present (Figure 6).

**Figure 5 materials-15-07529-f005:**
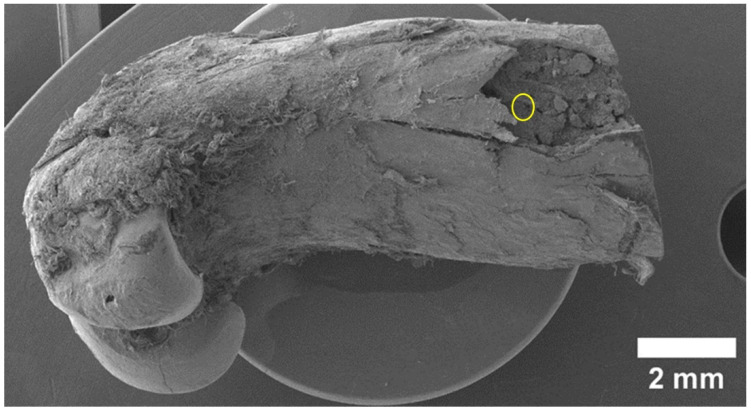
SEM image of sample after 6 months.

**Figure 6 materials-15-07529-f006:**
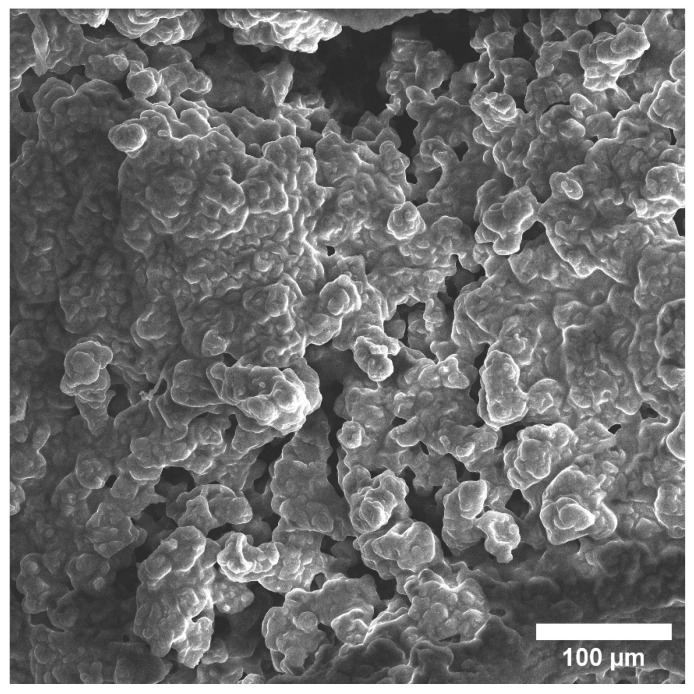
Enlarged SEM image of spherical discharge at implant site after 6 months.

### 3.2. EDX Analysis

Without taking gold into account (because it was deposited on the surface of the samples), oxygen, phosphorus, carbon and magnesium (and, to a lesser extent, calcium) form a large proportion (in wt.%) of the released materials (Figure 7, Table 1).

After 1 month, an oxygen-containing area with organic compounds (C) formed around the implant. However, the surface topography shows that the compounds, including C, Ca, O, and P, are closer to the bone, and Mg is present in relatively small amounts compared with deeper areas. C, Ca, O, P, and Mg are located in approximately the same areas. Therefore, we traced initial-stage osteoconduction within a month in one bone sample with an implant installed.

This image shows discharges close to a spherical shape, which are located relative to the implant closest to the bone (Figure 8). According to the composition, EDX found that they mainly consist of Mg, O, and P, and to a much lesser extent Na and C. With a larger increase in these spherical precipitates, it becomes clear that they consist of many sharp plates (Figure 9). At a magnification of 20 microns, we visualized a spiral structure, which consisted of many crystals of a certain shape (presumably HA crystals) growing from a common center and forming a cellular structure and certain relief, which were layering on each other. We hypothesize that under these conditions, in the interval from 1 to 3 months, a new bone is formed, and the structure of the new bone formation is cellular, consisting of fractal clusters of a certain spiral device.

When studying the chemical composition, it becomes clear that they consist of inclusions of Mg, O, P and, to a lesser extent, Na. These crystals signal the passage of magnesium bioresorption, forming an interface from compounds with oxygen and phosphorus (probably from interaction with hydroxyapatite and calcium phosphate—the main mineral components of bone tissue). However, this process is incomplete, and the elemental distribution is uneven and represented by different morphologies.

After 3 months, we observed pores and rounded organic discharges at the intended installation site. However, when examining the surface using EDX, we found no traces of the Mg compound in three areas of the bone. All three sites contain: C, O, Na, P, Ca, and Au (Figure 10). We also found Al, Fe, and S in extremely small volumes (<1 wt. %). We did not detect Mg, which leads us to conclude that it is not present on the surface of this 3-month-old bone sample implant. It was likely completely resorbed and replaced by bone tissue (Table 2).

We found similar results with a bone sample at 6 months. There are no visible traces of the implant at the implant site, and only spherical-shaped organic discharges are present. We did not find Mg in the elemental distribution map and, consequently, its compounds were also absent. Again, we detected C, O, Na, P, Ca, and Au, among which the highest in weight percent are C and O. These indicators possibly indicate the standard mineral components of bone—hydroxyapatite (Ca_5_(PO_4_)_3_(OH)) and calcium phosphate (Ca_3_(PO_4_)_2_) (Figure 11). Moreover, the elemental distribution relative to previous samples is quite uniform, which may indicate a completely replaced and homogeneous bone tissue (Table 3). However, the data differ in the literature, and further experiments are required [10].

## 4. Discussion

Several million people suffer annually from bone fractures caused by accidents (car accidents, industrial disasters, etc.) or various diseases [26]. The treatment of such patients is impossible without the use of fixation structures, such as plates, screws, and meshes [27,28]. In Russia, on average, 400,000 operations are performed per year using metal structures [28]. The financial volume of the global market for fracture fixation systems is more than USD 5 billion [29]. However, using titanium systems has a number disadvantages, such as a limitation of bone growth, which is extremely important in pediatric practice; problems associated with radiation therapy (oncology treatment); tactile sensitivity of the plate; and the limitation of limb movement [30,31,32,33,34]. In 2018, more than 170,000 operations were performed in Germany to remove titanium structures [35]. In Europe and the USA, 80% of patients undergo removal of metal structures after osteosynthesis [36,37,38]. In Germany in 2007, the estimated yearly costs of these procedures exceeded EUR 430 million, and in Russia they amounted to about RUB 6 billion [31,38]. Reducing the number of such operations will benefit the patients themselves, as well as reducing the financial burden on the global healthcare system.

Due to their mechanical and biocompatible properties, Mg-based connectors represent a promising technology compared with conventional materials used for the manufacture of plates and screws for osteosynthesis [39,40]. Mg-based alloys have the ability to decompose under physiological conditions and demonstrate compatibility with living tissues without any toxic, destructive, or negative immunological reactions [41,42,43,44,45]. Mg-based alloys promoted the formation of bone tissue during in vivo experiments on large and small animals [8,46]. However, the biodegradation of magnesium alloys leads to the release of gaseous hydrogen, which, if not controlled, can have a negative impact on regeneration processes [47,48,49]. The high degradation rate, uneven distribution of resorption, and localized corrosion of magnesium alloys may hinder the further development of this technology [50,51].

Recently, researchers have investigated the addition of Zn to the alloy, which can correct the rate and volume of hydrogen released. The results of studies have shown that Mg-Zn alloys have excellent mechanical properties, biocompatibility, and higher corrosion resistance [52]. Additionally, the addition of Zn to Mg alloys can significantly reduce H2 emission [53,54]. However, depending on the zinc content in binary Mg–Zn alloys and phase distribution, the corrosion resistance of Mg–Zn alloys varies greatly. Zhang et al. implanted Mg-6Zn alloy rods into the body of rabbits, and their results, which were obtained by the weight loss method, showed that Mg alloy can be gradually absorbed in vivo with a decomposition rate of 2.32 mm/year without heart, liver, kidney, and spleen disorders. In addition, six weeks after implantation, the subcutaneous H2 gas that accumulated as a result of alloy decomposition disappeared without noticeable adverse effects [55].

Our SEM and EDX study results highlight that the corrosion front is composed of calcium and phosphate, most likely in the form of an oxide layer. Similar results are described in Su Y. et al. [56]. The presence of these ions can lead to conversion, which can be beneficial to the bone regeneration process as the pins are optimally integrated into the bone matrix.

According to our study results regarding the area of previously installed pins, we determined a regenerate containing calcium phosphate and, presumably, hydroxyapatite. Additionally, some researchers also believe that the formation of more complex Mg_3_(PO_4_)_2_ is possible [27,29]. However, such data are mainly described with the subcutaneous injection of a rod based on magnesium alloys. When studying the chemical composition of the interface on the element distribution map, we found Zn only 1 month after implantation. Klima et al. also studied the chemical composition of the implant–bone interface using SEM-EDS and found a thin and compact phosphate-based layer (3–5 μm) on the surface of Zn-based implanted screws, regardless of the implantation period [46]. They found calcium and zinc in this layer, which led them to suggest the formation of a complex or mixture of Ca/Zn phosphates under in vivo conditions. They explained the formation of such a layer by the interaction of Zn^2+^ ions released upon dissolution of the experimental alloy with body fluids containing Ca^2+^, HXPO4(3–x)–, and other ions. These formed phosphates are subsequently deposited on the implant surface [56,57]. The formation and precipitation of these phosphates is also affected by the presence of proteins that can complex zinc and ions from solutions. These complex compounds subsequently enhanced phosphate precipitation [58,59]. Kubasek J. et al. discovered that a phosphate-based layer is formed at an early stage (up to several days) after implantation [18,60]. The formation of a phosphate layer is beneficial for further bioresorption. According to Su et al., the surface layer of zinc phosphate (ZnP) increases the cyto- and hemocompatibility of Zn-based materials [56]. They also reported increased antibacterial activity in ZnP-coated samples. Chou et al. studied the effects of ZnP coatings on the behavior of organic bone regeneration (GBR) membranes and described the antibacterial activity of zinc phosphate [61]. Thus, the formation of a ZnP layer with predictable antibacterial activity, which was observed by Klima et al., may be the reason why only a very limited inflammatory response has been recorded [46]. In our study, the nail was in close contact with both the cancellous bone and cortical layer. In the dynamics of our experiment, and especially at the post-six-months stage, young bone-tissue formation in the osteotomy zone is visualized, which indicates the complete biodegradation of the installed pins.

Furthermore, it is impossible to detect residual Mg by EDS surface scanning if it diffused into the bone to a depth greater than a few microns, and we could not find residual Mg on the specimen surface after 3 and 6 months of implantation.

Based on our results, we believe that the tested biomaterials can be used in a clinical setting without causing side effects. However, it is necessary to further increase the number of observations and conduct additional clinical studies using full-fledged structures for osteosynthesis, as well as using additive technologies for 3D printing individual products. Moreover, our plan is to investigate our hypothesis using new samples as follows:

(a) Prepare specimens with implanted Mg-based pins at 1-month steps until 6 months;

(b) Cut, grind, and polish the cross-section of each specimen;

(c) Perform final delicate polishing with Ga-FIB in the SEM chamber to remove the oxide layer;

(d) Reveal Mg bone diffusion by combining FIB-EDS and FIB-TOF-SIMS methods.

## 5. Conclusions

In our study, we successfully implanted Mg-Zn-Ga alloy pins in 12 Wistar rats, which we followed up for 6 months with sampling at 1, 3, and 6 months. The alloy is completely resorbed within a period of 1 to 6 months. We determined a spiral structure when analyzing the “bone–implant” interface. This structure is presumably similar to hydroxyapatite crystals. In the future, it will be necessary to conduct additional research on a larger number of animals and the use of finished products. Our obtained results will make it possible to create the most effective types of fixing structures consisting of bioneutral and low-toxic elements made of bioresorbable metals for various branches of medicine, making it possible to avoid repeated surgical intervention in the future.

## Figures and Tables

**Figure 1 materials-15-07529-f001:**
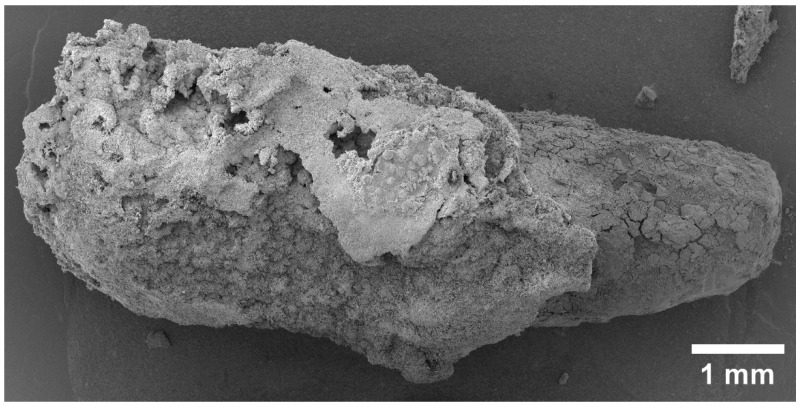
SEM image of magnesium implant surface obtained by reflection of secondary electrons 1 month after installation.

**Figure 2 materials-15-07529-f002:**
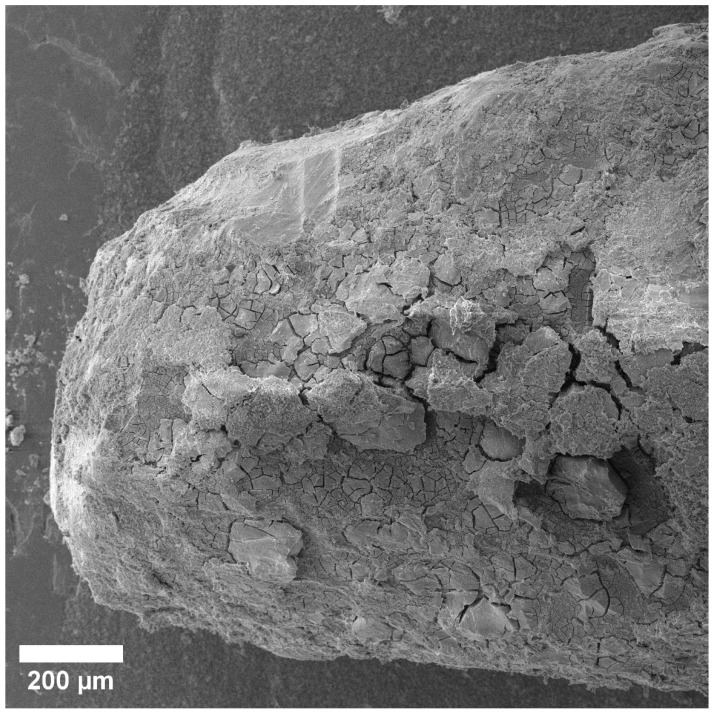
Close-up SEM image of implant site.

**Figure 7 materials-15-07529-f007:**
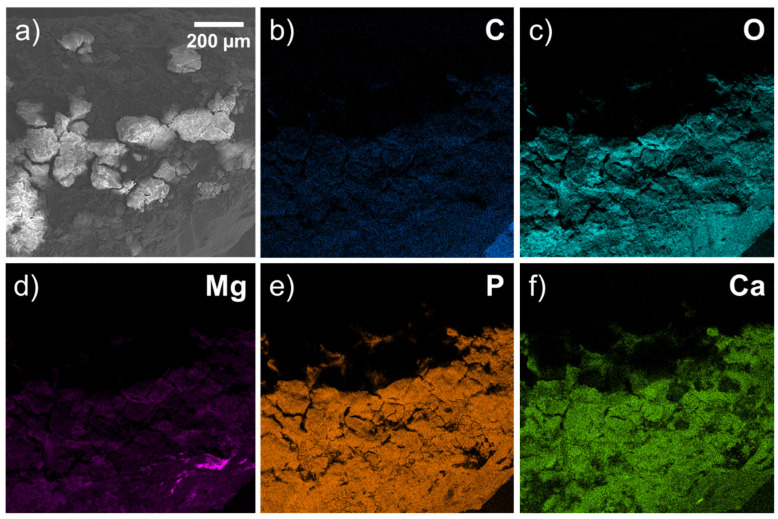
EDX image of surface relief of implant site after 1 month: (**a**) SEM image of bone surface; (**b**–**f**) elemental maps.

**Figure 8 materials-15-07529-f008:**
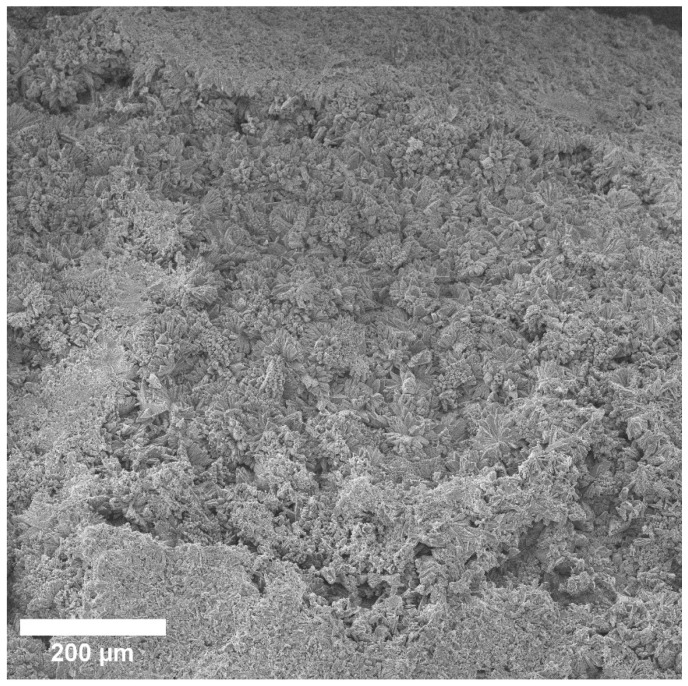
Spherical discharges located on site of bone implant.

**Figure 9 materials-15-07529-f009:**
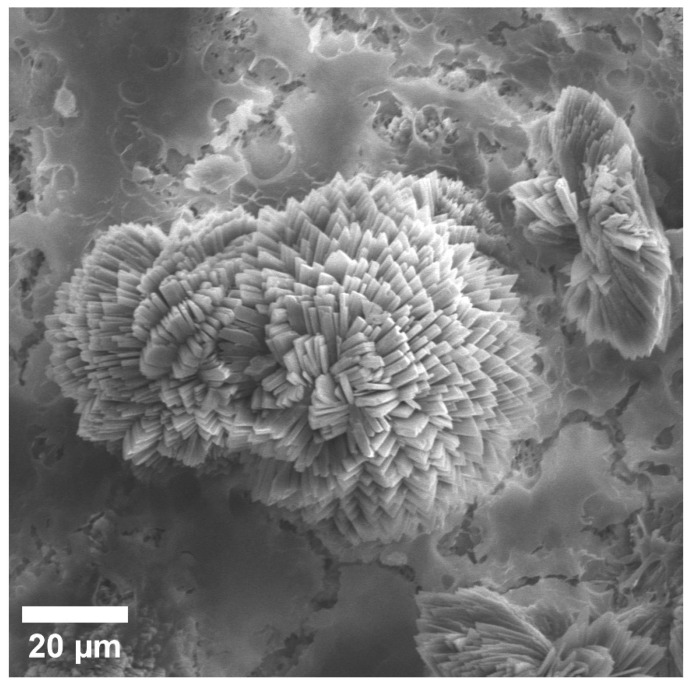
SEM image of multiple lamellae making up spherical highlights.

**Figure 10 materials-15-07529-f010:**
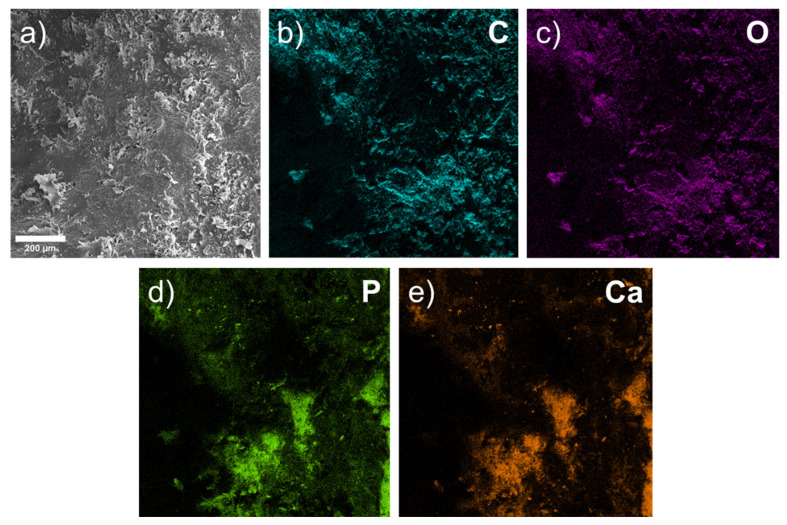
Layered image of EDX sample after 3 months: (**a**) SEM image of bone surface; (**b**–**e**) elemental maps.

**Figure 11 materials-15-07529-f011:**
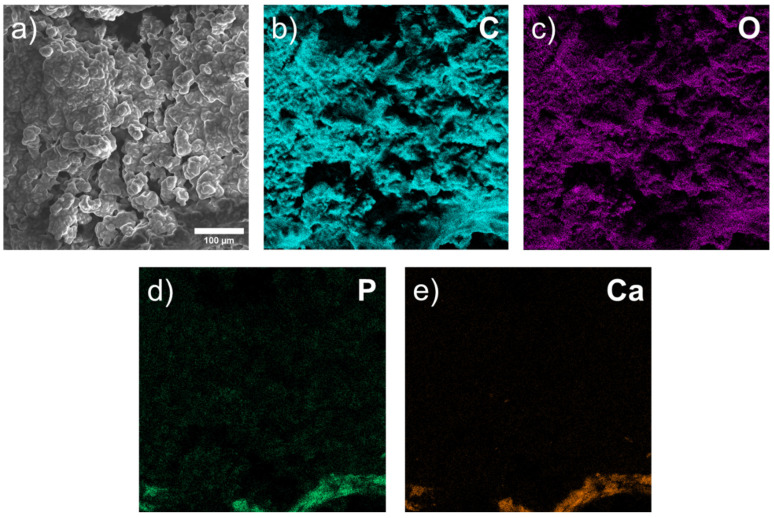
Layered image of EDX sample after 6 months: (**a**) SEM image of bone surface; (**b**–**e**) elemental maps.

**Table 1 materials-15-07529-t001:** Total spectrum of map and distribution of elements after 1 month.

Element	Line Type	Weight %	Sigma Weight %	Atom %
C	K-series	15.58	0.09	25.60
O	K-series	41.46	0.06	51.14
Mg	K-series	11.80	0.02	9.58
P	K-series	13.46	0.02	8.58
Ca	K-series	5.32	0.01	2.62

**Table 2 materials-15-07529-t002:** Total spectrum of map and distribution of elements after 3 months.

Element	Line Type	Weight %	Sigma Weight %	Atom %
C	K-series	60.09	0.07	82.36
O	K-series	12.35	0.05	12.71
P	K-series	1.95	0.01	1.04
Ca	K-series	3.93	0.01	1.62

**Table 3 materials-15-07529-t003:** Total spectrum of map and distribution of elements after 6 months.

Element	Line Type	Weight %	Sigma Weight %	Atom %
C	K-series	78.78	0.05	86.39
O	K-series	15.83	0.05	13.03
P	K-series	0.23	0.01	0.10
Ca	K-series	0.35	0.00	0.12

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
