# Peer review of "Bone Remodeling Interaction with Magnesium Alloy Implants Studied by SEM and EDX"

_materials, 2022, doi:10.3390/ma15217529_

Round 1

Reviewer 1 Report

The author made an endeavor to figure out the biodegradation level of Mg-2Zn-2Gd alloy by using the SEM and EDS. This paper well demonstrated that the bioresorption of Mg-2Zn-2Gd alloy according to the time after surgery. Despite the micrographs, the reviewer has several concerns that require major revisions.

 1.       The reviewer concerned about the quantitative results from EDS mapping analysis on the 3-dimensional surface topography. As the author already mentioned in the paper, the secondary electrons are strongly affected to the surface topology. For example, if the residual Mg implants would not be detected by secondary electrons if they located deeper area under the discharge or layers.

2.        In addition, there are not enough explanations for all the micrographs. For instance, the EDS mapping results are ambiguous to distinguish between elements.

3.       If there is a result of comparable reference material, it is considered the significance of SEM as a tool to evaluate the bioresorption can be given.

Reviewer 2 Report

Journal: Materials (ISSN 1996-1944)

Manuscript ID: materials-1913628

Type: Article

Title: Energy dispersive X-ray spectroscopy and scanning electron microscopy as tools for evaluating the bioresorption of articles made of magnesium alloys (experimental study)

Recommendation: Reject

In this paper, the preparation of the Mg-2Zn-2Ga alloy was made, the production of pins and the analysis of their biodegradation level based on SEM and EDX after an experimental study on rats were carried out, which provides an interesting contribution on the clinical applications of this type of Mg based alloys.

However, it was not sufficient deeply. Authors only report their results and discussed superficial about them, and did not make some definite conclusions. Hence, it is not worth publishing, and presents a number of shortcomings that must be addressed, as listed below:

1.      In the “abstract” section, the first occurrence of the abbreviations (“SEM”, “EDX”); should be explained.

2.      In lines 46-50, it was not stated that the subjects of these studies were magnesium alloys. And please list several literatures on the modification of magnesium alloys of significant researches.

3.      In line 53, the conjunctive adverb “however” may be a misnomer. It is strongly advised that the authors take all necessary measures to improve language quality and style.

4.      In the “introduction” section, authors should replace some old references reasonably by the latest literatures to introduce the state-of-the-art advances in Mg-Zn-Ga alloy in the revised manuscript.

5.      In lines 116-119, “All images……appearance of samples” is recommended to put it in Section 2.3, and avoid spending a lot of time talking about non-experimental results in the “results” section.

6.      In line 133, for the “After 3 months, it is difficult to determine the location of the magnesium implant“, does that mean the implant dissolved? Or did the implant come loose and fall? Please explain it in detail.

7.      It is recommended to add arrow or other marks in the pictures, which facilitates readers to identify the pictures. For example, mark pores and rounded in Figure 3, indicate which morphology is organic segregations in Figure 4, etc. Please indicate in Figure 5 the enlarged area to which Figure 6 belongs. The title of Figure 4 is too simple, and it is recommended to modify it appropriately.

8.      Are the “organic discharges of a spherical shape” and “spherical discharge” in the results the same thing? In this paper, the expression form is recommended to be unified.

9.      The tables are not standardized, it is recommended to use the three-line tables.

10.  In Table 1, where is the “Al” from? In line 181, where are the “Al, Fe, S” from? Please explain them in detail.

11.  In lines 194-195, why can “hydroxyapatite (Ca5(PO4)3(OH)) and calcium phosphate (Ca3(PO4)2)” be confirmed? Is it just speculated because they are the main components of bone tissue?

12.  In the “discussion” section (lines 203-215), “Magnesium (Mg) is……this technology [29, 30].” should be placed in the “introduction” section as appropriate; Since there are no in-depth discussions, the rest of the parts (“As part of our……individual products.”) can be integrated with the “results” section; It is recommended to write a separate section (Conclusions), which briefly and clearly describes the conclusions of this paper.

Author Response

Dear Editor and Reviewers,

We are grateful for the Reviewers’ valuable comments concerning the manuscript, “Bone remodelling interaction with magnesium alloy implants studied by SEM and EDX”. In the light of the questions and recommendations received we revised the manuscript extensively from title to grammar to content. Please find below our responses to the Reviewers’ comments one by one. We feel that the manuscript has been significantly improved and hope that the corrections meet the requirements for publication.

With kind regards,

Nikolay Redko

Original title:

“Energy dispersive X-ray spectroscopy and scanning electron microscopy as tools for evaluating the bioresorption of articles made of magnesium alloys (experimental study)”

Revised title:

“Bone remodelling interaction with magnesium alloy implants studied by SEM and EDX”

Reviewer #2

In this paper, the preparation of the Mg-2Zn-2Ga alloy was made, the production of pins and the analysis of their biodegradation level based on SEM and EDX after an experimental study on rats were carried out, which provides an interesting contribution on the clinical applications of this type of Mg based alloys. However, it was not sufficient deeply. Authors only report their results and discussed superficial about them, and did not make some definite conclusions. Hence, it is not worth publishing, and presents a number of shortcomings that must be addressed, as listed below:

  1. In the “abstract” section, the first occurrence of the abbreviations (“SEM”, “EDX”); should be explained.

Thank you. The indicated abbreviations were explained as advised.

  1. In lines 46-50, it was not stated that the subjects of these studies were magnesium alloys. And please list several literatures on the modification of magnesium alloys of significant researches.

Thank you for the detailed comment. We stated that information provided in lines 46-50 is about magnesium alloys. Also several literatures on the alloying and modification of biodegradable magnesium alloys were added:

Amukarimi, S.; Mozafari, M. Biodegradable Magnesium Biomaterials—Road to the Clinic. Bioengineering 2022, 9, 107, doi:10.3390/bioengineering9030107.

Sharma, S.K.; Saxena, K.K.; Malik, V.; Mohammed, K.A.; Prakash, C.; Buddhi, D.; Dixit, S. Significance of Alloying Elements on the Mechanical Characteristics of Mg-Based Materials for Biomedical Applications. Crystals 2022, 12, 1138, doi:10.3390/cryst12081138.

Jung, O.; Hesse, B.; Stojanovic, S.; Seim, C.; Weitkamp, T.; Batinic, M.; Goerke, O.; Kačarević, Ž.P.; Rider, P.; Najman, S.; et al. Biocompatibility Analyses of HF-Passivated Magnesium Screws for Guided Bone Regeneration (GBR). IJMS 2021, 22, 12567, doi:10.3390/ijms222212567.

Chen, J.; Xu, Y.; Kolawole, S.K.; Wang, J.; Su, X.; Tan, L.; Yang, K. Systems, Properties, Surface Modification and Applications of Biodegradable Magnesium-Based Alloys: A Review. Materials 2022, 15, 5031, doi:10.3390/ma15145031.

  1. In line 53, the conjunctive adverb “however” may be a misnomer. It is strongly advised that the authors take all necessary measures to improve language quality and style.

Thank you for the comment. The text of the manuscript was significantly rewritten while the grammar and punctuation were double-checked.

  1. In the “introduction” section, authors should replace some old references reasonably by the latest literatures to introduce the state-of-the-art advances in Mg-Zn-Ga alloy in the revised manuscript.

Thank you for the valuable comment. The reference list was updated. The old references were replaced by new ones.

  1. In lines 116-119, “All images……appearance of samples” is recommended to put it in Section 2.3, and avoid spending a lot of time talking about non-experimental results in the “results” section.

Thank you. This sentence was moved into the Section 2.3 as suggested.

  1. In line 133, for the “After 3 months, it is difficult to determine the location of the magnesium implant“, does that mean the implant dissolved? Or did the implant come loose and fall? Please explain it in detail.

Yes, physical bioresorption of the implant occurred after 3 months, and it was impossible to visualize it after this period. The implant remained within the bone all this time and no loss of stability was found.

  1. It is recommended to add arrow or other marks in the pictures, which facilitates readers to identify the pictures. For example, mark pores and rounded in Figure 3, indicate which morphology is organic segregations in Figure 4, etc. Please indicate in Figure 5 the enlarged area to which Figure 6 belongs. The title of Figure 4 is too simple, and it is recommended to modify it appropriately.

Thank you for the detailed comment. We made the drawings more visual as advised, namely, displayed important elements with the help of arrows and labels. Figure 4 title had been changed.

  1. Are the “organic discharges of a spherical shape” and “spherical discharge” in the results the same thing? In this paper, the expression form is recommended to be unified.

Right. These names are the same. We chose the term ‘spherical discharge’.

  1. The tables are not standardized, it is recommended to use the three-line tables.

Thank you. The style of tables was corrected according to the journal rules and guidelines.

  1. In Table 1, where is the “Al” from? In line 181, where are the “Al, Fe, S” from? Please explain them in detail.

Thank you for the remark. Indeed, it is our mistake to report all detected elements without checking. The presence of Fe and S is reported incorrectly and was added accidentally (now removed). The appearance of Al during EDS measurement is parasitic and has a well-established reason: it is connected with the presence of Al alloy components inside the SEM chamber; for instance, the specimen table is made from an Al alloy and may contribute to the signal through parasitic excitation. Now, all extraneous elements such as Fe, S, Al and Au (the sputtering material to reduce sample charging) were removed from tables and maps.

  1. In lines 194-195, why can “hydroxyapatite (Ca5(PO4)3(OH)) and calcium phosphate (Ca3(PO4)2)” be confirmed? Is it just speculated because they are the main components of bone tissue?

This is a hypothesis that can be confirmed by the two considerations as follows:

  1. a) the formation of crystal clusters on the bone surface.
  2. b) the presence of a spiral structure consisting of many crystals of sharp habit (external appearance) typical of crystalline phase and not possible for amorphous material.

This leads us to assume that under the conditions of experiment, in the interval from 1 to 3 months, new bone tissue was formed containing HAp nanocrystals.

  1. In the “discussion” section (lines 203-215), “Magnesium (Mg) is……this technology [29, 30].” should be placed in the “introduction” section as appropriate; Since there are no in-depth discussions, the rest of the parts (“As part of our……individual products.”) can be integrated with the “results” section; It is recommended to write a separate section (Conclusions), which briefly and clearly describes the conclusions of this paper.

Thank you. The Conclusions section was added where we highlighted the novelty of the study, discussed obtained results and plan new set of experiments to deeper understand phenomena of Mg-based implant bioresorption that are taken via dissolution and diffusion into the bone.

Reviewer 3 Report

1- The English language of the manuscript needs to be improved.

2- In the title, the authors stated the term “bioresorption” which means that the materials are removed by body fluid....is it appropriate in the title?

3- Figures 7,8,10 and 11 need to improve

4- Authors have to compare their results with other works.

5- Authors have to discuss the results more and more.

Round 2

Reviewer 1 Report

The reviewer appreciates the kind response from the author. The paper can be accepted without any further changes.

Author Response

Dear Reviewer!

Thank you for your time and appreciation of our manuscript!

Thank you for approving our publication!

Reviewer 2 Report

Accept

Author Response

(The authors gave the same response as above.)
